# Augmenting complex and dynamic performance through mindfulness-based cognitive training: An evaluation of training adherence, trait mindfulness, personality and resting-state EEG

**Chloe A. Dziego**[1]*, **Ina Bornkessel-Schlesewsky**[1], **Matthias Schlesewsky**[1], **Ruchi Sinha**[2], **Maarten A. Immink**[1,3], **Zachariah R. Cross**[1,4]

**1** Cognitive Neuroscience Laboratory, Australian Research Centre for Interactive and Virtual Environments, University of South Australia, Adelaide, Australia, **2** Centre for Workplace Excellence, University of South Australia, Adelaide, South Australia, **3** Sport, Health, Activity, Performance and Exercise (SHAPE) Research Centre, Flinders University, Adelaide, Australia, **4** Department of Medical Social Sciences, Northwestern Feinberg School of Medicine, Chicago, Illinois, United States of America

☯ These authors contributed equally to this work.
* Chloe.Dziego@mymail.unisa.edu.au

**Data Availability Statement:** All data and code is available on an Open Science Framework (OSF)

## Abstract

Human performance applications of mindfulness-based training have demonstrated its utility in enhancing cognitive functioning. Previous studies have illustrated how these interventions can improve performance on traditional cognitive tests, however, little investigation has explored the extent to which mindfulness-based training can optimise performance in more dynamic and complex contexts. Further, from a neuroscientific perspective, the underlying mechanisms responsible for performance enhancements remain largely undescribed. With this in mind, the following study aimed to investigate how a short-term mindfulness intervention (one week) augments performance on a dynamic and complex task (target motion analyst task; TMA) in young, healthy adults ($n = 40$, age $^{range} = 18$–38). Linear mixed effect modelling revealed that increased adherence to the web-based mindfulness-based training regime (ranging from 0–21 sessions) was associated with improved performance in the second testing session of the TMA task, controlling for baseline performance. Analyses of resting-state electroencephalographic (EEG) metrics demonstrated no change across testing sessions. Investigations of additional individual factors demonstrated that enhancements associated with training adherence remained relatively consistent across varying levels of participants' resting-state EEG metrics, personality measures (i.e., trait mindfulness, neuroticism, conscientiousness), self-reported enjoyment and timing of intervention adherence. Our results thus indicate that mindfulness-based cognitive training leads to performance enhancements in distantly related tasks, irrespective of several individual differences. We also revealed nuances in the magnitude of cognitive enhancements contingent on the timing of adherence, regardless of total volume of training. Overall, our findings suggest that mindfulness-based training could be used in a myriad of settings to elicit transferable performance enhancements.

repository (titled: Augmenting complex and dynamic performance through mindfulness-based cognitive training: an evaluation of training adherence, trait mindfulness, personality and resting-state EEG), accessible through the following link: https://osf.io/y9hvt/.

**Funding:** Preparation of this work was supported by a grant from the Defence Science and Technology Group [Research Agreement 9208] under the Research Network for Undersea Decision Superiority (RN-UDS). The funders had no role in study design, data collection and analysis, decision to publish, or preparation of the manuscript. IB-S was supported by an Australian Research Council Future Fellowship (FT160100437).

**Competing interests:** The authors have declared that no competing interests exist.

## Introduction

In Western psychology, mindfulness is commonly defined as a non-judgemental awareness and acceptance of the present moment [1]. It is formally induced by a variety of techniques, including sitting meditations, yoga movements and body scans [2]. Such practices are becoming more popular in clinical settings, as the benefits to wellbeing and mental health are increasingly being demonstrated in the literature [3,4]. However, in addition to its well-established benefits for mental health, human performance research indicates how mindfulness techniques can further improve cognitive functioning [5–8]. In this domain, engaging in mindfulness practices have been associated with enhancements in attentional processes [7,9,10], executive functions [11,12], working memory ability [12–14] and even creative thinking [15,16]. Despite demonstrating promising results, these studies are limited by the narrow nature of cognitive tasks employed (i.e., Attentional Network Task or simple motor sequence tasks [10,17,18]), which fail to explore how mindfulness may enhance performance in more naturalistic environments. Secondly, mindfulness regimes are typically of considerable duration [5] and investigations largely employ a between-groups approach, ignoring the individual variability in intervention outcomes [19]. Lastly, while frameworks have been developed regarding the underlying theoretical (e.g. through the fractioned model of executive functioning [11,20]) and neurobiological mechanisms of mindfulness (i.e., changes to anterior cingulate cortex, functional connectivity and inhibitory neurotransmitters [21,22]), there has been little investigation into how mindfulness may relate to long-term neural changes in resting-state electrophysiology. The current study thus aimed to implement a short-term mindfulness intervention to investigate performance enhancements in more multidimensional and dynamic settings (i.e., a medium-fidelity submarine simulation; the Target Motion Analysis task [23]), while also recognising the role of individual factors (i.e., differences in adherence, trait mindfulness, personality and resting-state neurophysiology).

### Mindfulness and cognitive function

Although mindfulness interventions have predominantly been employed within the realm of well-being and mental health, recent research has started to reveal its broader effects on cognitive functioning and performance [5–8]. Notably, performance enhancements have been demonstrated in a range of traditional cognitive tests including the Attentional Network Test [10,17], the Symbol Digit Modalities Test [24], Stroop Test [25] and in sequenced motor tasks [18,26]. These mindfulness interventions also do not need to be extensive or lengthy to produce significant results, highlighting their propensity to be an easily accessible cognitive training tool. Even engaging in very brief interventions (20 minutes each day for four days) improve visuo-spatial processing, working memory and executive functioning [24] and a single session of focused attention monitoring immediately prior to learning can improve consolidation of motor patterns [18,27]. Similar brief pre-test engagement in mindfulness techniques have reduced inattentional blindness [28], improved Stroop Test scores [29], increased flanker task performance [30] and working memory ability [31]. Overall, reviews in this area have concluded that practicing mindfulness is associated with gains in the domains of executive functioning [5,32,33] and working memory [34] that underlie generalised and transferable improvements in other cognitive functions. From a theoretical perspective, these enhancements are thought to arise from mindfulness practice's influence on two domains of executive attention: the inhibition of irrelevant information and improved attentional shifting [11]. Even with large heterogeneity in interventions, engaging in some type of mindfulness practice (whether short-term, longer term or single session modalities) appears to be beneficial to cognitive functioning.

Nevertheless, reviews of this literature have also reported non-significant effects of mindfulness training for attention and several forms of memory [8,33], while cautioning the lack of high quality evidence available [5]. Previous studies are further limited by their examination of mindfulness-based cognitive enhancements using traditional laboratory tasks, which do not address the more complex and dynamic nature of optimal performance in the real-world. Moreover, we note that the extant literature typically employs a between-groups approach, with a common prescribed number of sessions or session duration across the experimental group. These studies fail to capture the potential effect of volume of mindfulness training, as well as how individual factors may influence adherence to the training regime or reported performance outcomes. In addition to this, the timing of adherence or engagement in cognitive training sessions may play a role in derived benefits; that is, whether sessions are performed closer to the pre or post-intervention testing sessions. Here, such investigation could help to differentiate whether mindfulness-based training might be more beneficial as a 'consolidation' practice (e.g., helps to improve memory of the task instructions from the first session, theorised by Brown et al. [35]) versus an attentional/state enhancing tool (e.g., helps to create an optimal attention state for task performance in the second session; as demonstrated by single session, brief training enhancements [18,28,31,36]). The consideration of this individual variability in adherence, alongside participant characteristics, may help to explain the inconsistencies in previously reported results.

## Individual factors

A recent review by Tang and Braver [19] underscores the importance of adopting this individual differences perspective in mindfulness training research to help elucidate incongruencies in past studies. Here, they emphasise how dispositional traits are commonly reported moderators of intervention effects within clinical psychotherapies [37,38] and thus also require consideration within the investigation of mindfulness-based regimes.

In particular, levels of dispositional (or trait) mindfulness refer to intra-individual differences in the frequency with which individuals experience subjective awareness to the present moment (i.e., frequency of 'mindful states' [39], which has been shown to interact with outcomes of mindfulness-based interventions in depression [40] and within cognitive testing contexts [41]. Moreover, trait mindfulness has been shown to positively correlate with self-reported attentional capacity [42], suggesting that it is a necessary consideration within investigations of cognitive performance.

Additionally, researchers have highlighted that personality traits, such as the Big 5 [43], should also be recognised within this context. Notably, conscientiousness and neuroticism are often associated with facets of mindfulness [19,44–46]. Neuroticism can act as a potential barrier to intervention adherence [47], but, Tang and Braver [19] also note that those with high baseline neuroticism may demonstrate a greater propensity for beneficial outcomes from mindfulness interventions. Conscientiousness, on the other hand, can be a facilitator of intervention adherence [48,49]. Consistently, both neuroticism and conscientiousness are also known to demonstrate correlations (negative and positive, respectively) with measures of dispositional mindfulness [44–46]. These studies suggest there are intricate interrelationships between these personality constructs and mindfulness, highlighting the need for their consideration within experimental contexts.

## Neurophysiology of mindfulness

In addition to trait mindfulness and personality variability, individual neural differences may also influence outcomes from mindfulness training interventions. The neuroscientific

exploration of mindfulness has primarily focused on the investigation of on-task (or evoked) neural dynamics, with previous literature demonstrating that practicing mindfulness meditation is most consistently associated with increases in power across both alpha and theta frequency bands (see [50,51] for review). However, emerging research is beginning to establish that brain activity at rest may provide insights into the brain's functional capacity and highlight individual differences in information processing and performance capabilities [52–54]. While resting-state activity is often explored in the context of neurological disorder or dysfunction (e.g., [55–59]), it has also been investigated in relation to peak cognitive performance [60–64] and thus may be able to provide insights into salutary neuroplastic changes from mindfulness practice.

Two resting-state characteristics of electroencephalography (EEG) recordings that have been associated with cognitive functioning and performance include aperiodic 1/$f$ activity [60,63–66] and the individual alpha frequency (IAF) [62,67,68]. IAF refers to the frequency (Hz) of maximal neural power (uV) within the alpha band range (7.5–12.5Hz [69]). IAF is typically estimated during an eyes-closed resting state and has been found to positively correlate with IQ [62], visuo-perceptual ability [68,70,71] and memory performance [67,72]. More recent findings are demonstrating that IAF might not simply represent enhanced global cognition, but may indicate individual differences in information processing strategies [73,74]. Previous work has also demonstrated that higher IAF at rest is associated with faster adaptation to complex and dynamic tasks [61]. While oscillatory analyses of EEG during meditation have commonly demonstrated increases in alpha power overall [50,75], studies of IAF in particular have shown that experienced meditators exhibit lower IAF during meditation [76,77]. However, whether these changes persist long-term (altering the resting-state) and how they relate to enduring cognitive benefits is unknown.

Secondly, aperiodic 1/$f$-like activity refers to scale-free brain dynamics that have traditionally been ignored or treated as a nuisance variable in electrophysiological analyses [60,78,79]. Aperiodic 1/$f$ activity presents as a distinguishable quasi-straight line that can be observed when the EEG signal is transformed into its power spectral density [60,78,80]. Steeper slopes in this line during rest have been associated with increased performance on object recognition tasks [64], working memory tasks [60] and visuomotor tasks [63], as well as faster adaptation to novel language input [81]. However, flatter 1/$f$ slopes have also been associated with increased performance in memory tasks [82] and during learning in more complex and dynamic settings [61,65]. More generally, aperiodic 1/$f$ activity is largely thought to reflect the excitation/inhibition balance within neural networks [83,84]. This notion has been supported by studies exploring the influence of alcohol [85], propofol [86], sleep [79,87] and psychiatric disorders [58,59,88] on the 1/f slope. In the context of cognitive performance, it is thought that an optimal balance between excitation and inhibition is associated with peak cognitive functioning [10,61,63].

From a physiological perspective, a review of the neurobiological correlates of meditation suggest that it largely influences inhibitory mechanisms in the brain [21,89,90]. Therefore, as aperiodic activity is thought to reflect the excitation/inhibition balance in the brain [83,84], it is possible that steeper 1/$f$ slopes may be observed long-term following brief/sustained mindfulness practice. Consistent with this notion, meditation interventions have been successfully employed in the prevention of age-related declines in cognition [91,92], suggesting that meditation may impede age-related flattening of the 1/$f$ slope [93]. However, other EEG studies during meditation suggest contrasting effects of meditation on the 1/$f$ slope, with global power increases across all frequency bands from pre-to-post meditation [94], significant differences in delta and gamma between experienced meditators and novices [95] and long-term practitioners of meditation exhibiting increased gamma power, when compared to those who are

inexperienced [96]. One study specifically aiming to investigate 1/*f* dynamics in meditation reported that experienced meditators showed steeper slopes during mindfulness practice compared to when resting [76]. These mixed results call for further examination of both in-task EEG dynamics of mindfulness practice, but also how these changes might persist long-term and directly relate to cognitive functioning. Additionally, individual variability in resting-state metrics could alternatively distinguish those with neural profiles most likely to optimally engage in mindfulness practices and derive cognitive benefits.

## The current study

In summary, previous work has demonstrated cognitive enhancement from mindfulness training. However, findings have not been entirely straight-forward, suggesting that unaccounted factors (e.g., levels of adherence and individual factors) might moderate these effects. The current study therefore aims to investigate adherence to mindfulness-based cognitive training and individual differences, alongside performance-based enhancements in cognition, in the effort to provide more clarity to the findings. Another issue in this literature is that demonstrated performance enhancements from mindfulness training have been limited to more one-dimensional cognitive tasks (i.e., Stroop Test, Trail Making Tests, Digit Span, Attentional Network Task [5,8,11,97]), which raises questions about the extent to which cognitive gains from mindfulness training transfer to multifaceted, mutable tasks that are more representative of real-world performance situations. As such, the current study aims to extend these findings and assess the transferability of this cognitive training regime to more complex and dynamic environments using the Target Motion Analysis (TMA) task from the Control Room Use Simulation Environment (CRUSE [23]). The TMA task mimics the computerised console of a target motion analyst (responsible for estimating the movement of surrounding ships) within a submarine control room. Here, individuals must integrate multiple sources of information, while prioritising and continuously monitoring surrounding vessels (contacts) to perform well. Given the evidence that even brief interventions have been shown to induce performance enhancements [18,24], the current study will implement a one-week mindfulness intervention. Furthermore, to explore the accessibility of the intervention, the experiment will involve participants who are new to mindfulness practice. Previous research has demonstrated that even individuals who are meditation naïve can display cognitive benefits from mindfulness techniques and exhibit changes in electrophysiology [18,24,51,94]. Here, we used adherence to the mindfulness intervention as a continuous dependent variable. In this case, the study hopes to better capture how true engagement in the mindfulness practices (not just group assignment; and often, varied compliance), relates to performance enhancements.

We hypothesised that (A) performance enhancements in the TMA task from the pre- to post-training sessions will be predicted by adherence to mindfulness sessions (whereby improvements positively correlate with adherence). Here, we expect to see that augmentations in performance will be differentially influenced by self-reported enjoyment and individual timing of adherence (i.e., whether density of training is completed closer to the first or second testing session), allowing us to theorise about underlying mechanisms of mindfulness enhancements to performance. As an additional exploratory analysis, motivated by findings in Brown et al. [35]—where post-training mindfulness expedited wakeful consolidation of motor sequences—we examined how adherence to the training protocol prior to sleep might influence its direct effects on performance, irrespective of total overall volume.

Secondly, we hypothesised that (B) individual factors (dispositional mindfulness, conscientiousness, neuroticism and resting-state neural metrics) will interact with benefits gained from the mindfulness intervention. In particular, we predicted that initial levels of dispositional

mindfulness would interact with benefits gained from the mindfulness-based cognitive training regime. Higher levels of neuroticism, and lower levels of conscientiousness, should be negatively related to adherence. Further, resting-state metrics may provide insight into individual capacity to optimally engage in mindfulness practices (and thus derive benefits), or could help to track neural changes (associated with improved performance) following mindfulness interventions. Thus, in line with majority of findings in the performance literature–alongside in meditation-related studies—we expect to observe steeper $1/f$ slopes in individuals, in comparison to pre-intervention measures, following the mindfulness intervention.

## Method

### Participants

Forty-two adults ($M_{age}$ = 23.43, range: 18–38, females = 28) volunteered to participate in the current study. To participate, individuals were screened and must have reported no history of neurological, cognitive, psychiatric or language disorders. All participants were right-handed, with normal to corrected-to-normal vision, and were not taking any medications that may affect EEG. Participants also had no experience with formal cognitive training or mindfulness programs. As to not bias participant recruitment and expectations, advertising materials did not reference mindfulness or meditation [98,99]. Ethics approval was acquired from the University of South Australia's ethics committee (project no. 204054) prior to the commencement of data collection. Recruitment ran from 19th August 2021 to 25th January 2023. Participants received an honorarium of AUD$120. Two participants' data were removed due to withdrawing from the study. Consequently, data from forty eligible participants were used in the final analysis ($M_{age}$ = 23.50, range: 18–38, females = 26).

### Mindfulness Attention Awareness Scale (MAAS)

Participants completed the Mindfulness Attention Awareness scale [39] through an online survey accessible through *Qualtrics* [100]. This scale includes 15-items that measure levels of dispositional (trait) mindfulness. Here, participants respond to how frequently they have experienced certain events using a 6-point Likert scale ("Almost Always" to "Almost Never"). For example, "I rush through activities without being really attentive to them". Individual MAAS scores are calculated by summing responses (where "Almost Always" = 0, and "Almost Never" = 6), with higher scores relating to higher levels of dispositional mindfulness.

### Personality questionnaire

To investigate how individual personality factors may play a role in outcomes of mindfulness interventions, participants completed a brief personality questionnaire on *Qualtrics* [100] prior to visiting the lab. Questions originated from the short-form International Personality Item Pool (IPIP; [101–103]) and aimed to target the two Big 5 [43] traits of interest (neuroticism and conscientiousness). Participants are prompted to view a list of statements "that capture who you are as a person." For each trait, 8 items are presented in the form of statements (e.g., "I like to follow rules strictly") where participants respond on a 5-point Likert scale from "Strongly Disagree" to "Strongly Agree". Individual participant scores for each trait are calculated by summing the responses (where "Strongly Agree" = 5 and "Strongly Disagree = 1) and reverse scoring respective items as necessary. Higher scores are related to higher levels of the specified trait. To ensure that participants responded to all items within the questionnaire, we enforced a force-choice paradigm, in which the next question was only presented once the prior section was answered.

## Mindfulness-based cognitive training intervention

A web-browser based online mindfulness-training program was implemented through short audio recordings voiced by an experienced mindfulness instructor (based on mindfulness techniques previously used in research [35,104,105]). Seven distinct audio tracks (approximately 5 minutes in length) were recorded for the current study. All transcripts and audio recordings can be accessed by the Open Science Framework (OSF) repository here: https://osf.io/y9hvt/. These recordings guided participants through a variety of focused attention and open monitoring mindfulness practices (for full descriptions of these techniques, see [2,106]), using cues such as "direct your attention to your breath". Audio recordings on day 1 and day 2 involved observing and focusing on breathing or the body, while day 3 expanded to sequentially moving attention across the entire body (in the style of focused attention meditations). Day 4, 5 and 6 comprised of focusing attention on observing and counting the breath but began to introduce concepts of open monitoring meditations (i.e., observing when distractions arise). Audio recordings for days 6 and 7 gradually instructed more open monitoring techniques, where attention was expanded to noticing thoughts and non-judgmentally experiencing external surroundings.

Participants were able to access the audio recordings through their web-browser directed to an online repository hosted on *GitHub Pages* (GitHub Inc., 2021; accessible at https://github.com/chloeadee/cognitivetraining). Each recording was accessed individually on a daily basis based on the layout of the webpage. HTML code for the webpage was produced using *RMarkdown* in R [107]. Instructions on the webpage encouraged participants to engage in the brief mindfulness sessions three times per day; preferably in separate sessions (i.e., three exposures to the same 5 minute recording; for a total of 15 minutes each day). Following their last session each day, the website prompted participants to complete a 'post-training' questionnaire about their experience and how many times they were able to partake in the sessions (see post-training survey below). Unbeknownst to participants, training adherence was additionally tracked using *Google Analytics* and *Google Tag Manager* (Google Inc., 2021). Tracking metrics recorded when and how many times participants accessed the site's various pages and completion of a mindfulness session was counted when audio tracks were played for > 90% of their duration. These objective metrics were used to identify individual participant adherence.

## Post-training survey

To assess how individual preferences for mindfulness may influence outcomes, participants were encouraged to complete a post-training survey following their final mindfulness session each day. This survey aimed to capture which day of training they had fulfilled (1–7) and how many sessions were completed (0–3), alongside measurements of how participants felt during their exercises. Here, participants were instructed to place a moving slider from 0 (not at all) to 100 (very) in response to statements about their mindfulness session. For example, items included "You felt calm during the exercise" and "You felt energised after the sessions." For the current study, we focused on item 6 ("You enjoyed the exercises") to explore how individual participant preference for the intervention may predict cognitive outcomes. For each participant, an average enjoyment score was calculated from the seven days of responses. We note that some participants did not always engage in their prescribed mindfulness session (or complete their surveys) and thus scores were created from any potential responses provided (range 1–9 responses; three participants provided more than one response per day, resulting in a maximum of 9 responses).

## Control-room-use-simulation environment

To investigate performance in a more dynamic setting, we used the dual-screen Target Motion Analysis (TMA) task from within the Control Room Use Simulation Environment (CRUSE) [23]. CRUSE is a medium-fidelity simulation involving multiple operator stations that mimic the roles of individual submariners. The aim of the TMA operator within CRUSE is to create a 'tactical picture' of the surrounding vessels (known as contacts) by integrating multiple sources of sensor information (see Fig 1 for a schematic of a participant engaging in the TMA task). Participants must combine information from SONAR, optronics and the Track Manager to develop a 'solution' (an estimate of location and movement) for each surrounding contact (for a detailed description of the task see [61]). Performance can be measured by the accuracy of plotted solutions (known as the tactical picture error; TPE). TPE is a measure of the distance between the coordinates of plotted solutions and the simulation's objective truth of the surrounding environment. The metric is weighted by a contact's priority (i.e., fishing vessel or warship), course, range and behaviour (see [23]) and is logged every twenty seconds. Here, lower TPE signifies better performance. The simulations default solutions were included in the

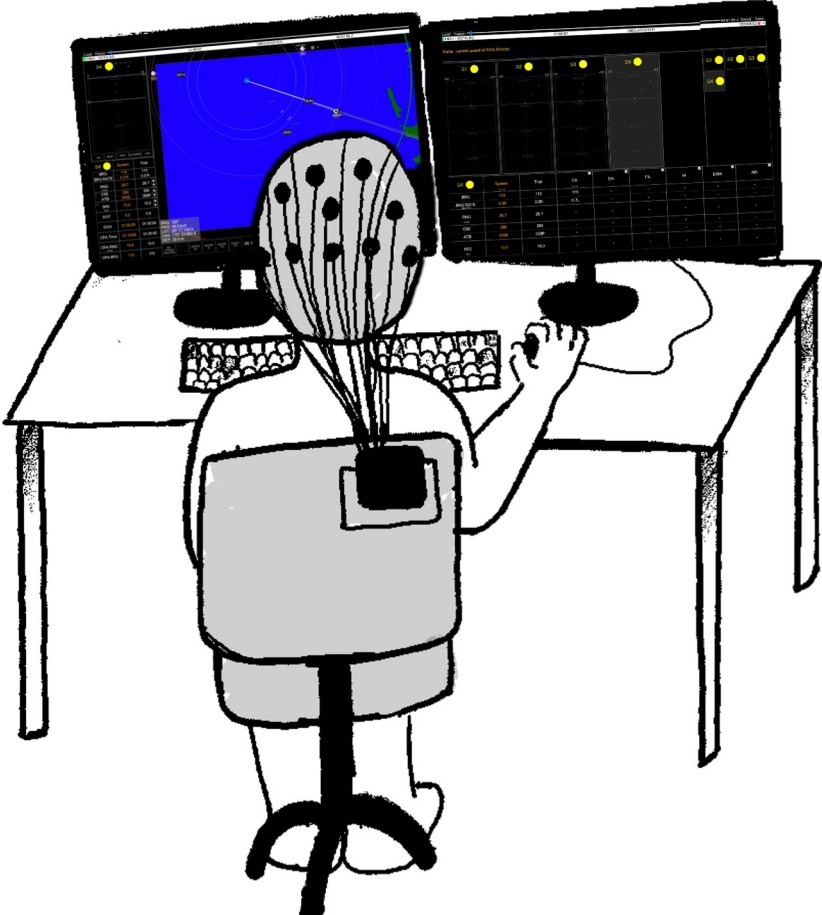

**Fig 1. Schematic of the TMA simulation. This figure depicts a participant** engaging in the dual-screen TMA task within CRUSE. On the left monitor, participants are presented with a geoplot of their submarine, their trial tote (for adjusting contact parameters and setting new solutions) and current plotted solutions. On the right monitor, participants can view the dot stacks and compare their parameters with information received from SONAR, to assess the accuracy of their current solutions. For a more detailed diagram of the TMA task, please see [61].

initial calculation of TPE (with all participants beginning with a score of 17,500). Three scenarios were created for current study; (1) a practice scenario (involving two contacts with simple movement), (2) a pre- and (3) a post-training scenario. Both pre- and post-training scenarios included five contacts (with more erratic movement) and were reflections and inversions of each other, for comparable difficulty.

### Protocol

Researchers were contacted by potential volunteers. After ensuring eligibility through an online screening process, participants were provided access to an educational video about TMA and a quiz to ensure they understood the general components of the task. Participants were then invited to the Cognitive Neuroscience Laboratory at the University of South Australia's Magill campus for a pre-intervention testing session. On the day of their visit, participants provided written consent to participate and completed two minutes of resting state EEG recordings with eyes closed, following their cap fitting. Participants then engaged in the practice scenario of the TMA task (approximately 30 minutes in length) where they were provided instructions by the researcher (for the practice session script, please see the S1 File). Here, participants could ask questions and clarify the instructions. Following the practice scenario, participants engaged in the 40-minute testing scenario (of increased difficulty) without researcher feedback.

Following their pre-intervention testing session, participants were provided access to the mindfulness-based training exercises through email. Participants were encouraged to participate in the exercises three times per day (for a total of 15 minutes each day). After seven days, participants were invited to a second testing session in the laboratory. Here, they were fitted with an EEG cap, completed a second resting-state recording (eyes closed; two minutes in length) and completed a subsequent 40-minute testing scenario of the TMA task. No further practice was included in the second in-lab visit. The order of presentation of the two testing scenarios was randomised across participants.

### Data analysis

#### EEG recording and pre-processing

Electrophysiological metrics were calculated from two minutes of EEG recorded at rest. EEG was recorded from 32 Ag/AgCl electrodes mounted in an elastic cap (actiCAP, Brain Products GmbH, Gilching, Germany). Eye blinks were monitored with electrode pairs placed above/ below the left eye. Channels were amplified using a LiveAmp 32 amplifier (Brain Products, GmbH) and sampled at a rate of 1000 Hz. Impedances were kept below 10kΩ. Pre-processing and analysis of EEG data was performed using MNE-python [108]. Raw data were band-pass-filtered from 1 to 40 Hz (zero-phase, hamming windowed finite impulse response) and re-referenced to the average of left and right mastoid electrodes (TP9/TP10). Data were also resampled to 250 Hz.

#### Individual alpha frequency & 1/*f* estimation

Individual alpha frequency (IAF) was calculated in line with procedures outlined in Corcoran et al. [109]. IAF was calculated from seven occipital-parietal electrodes (P3/P4/O1/Oz/O2/P7/ P8) using *philistine.mne.savgol_iaf* [110] in MNE-Python [108]. These electrodes were chosen as the individual alpha peak is most predominantly discernible across these areas. This method uses a Savitzky-Golay filter (frame length = 11 frequency bins, polynomial degree = 5 [111]) to smooth the power spectral density (PSD) before determining the peak of activity within the

defined frequency range (in this case, 7–13 Hz). Here, PSD estimates are gathered from fast Fourier-transformed EEG signals. This method calculates IAF using both Peak Alpha (PAF) and Centre of Gravity (CoG) techniques. For a full description of the technique, see [109].

Aperiodic 1/*f* activity was separated from oscillatory activity in resting-state recordings using the irregular-resampling auto-spectral analysis method (IRASA) [112] implemented in the YASA toolbox [113] in MNE-Python. This method distinguishes the aperiodic component of EEG signals from narrowband signals through a process of resampling, providing both a slope and intercept value. The slope and intercept refer to the respective metrics gathered from a regression line fitted to the estimated aperiodic signal (see Fig 2 for an example).

## Statistical analysis

*R* version 4.2.0 [107] was used to perform statistical analyses with the following packages: *tidyverse* v.1.3.1 [114], *lme4* v.1.1.29 [115], *car* v.3.0.13 [116], *splines* v. 4.0.2 [115]. Model fits were examined using the *performance* package v.0.9.0 [117]. *LmerOut* v.0.5.1 [118] was used to produce model output tables. To extract modelled effects and visualise data, *ggplot2* v.3.3.0 [119], *ggeffects* v.1.1.2 [120] and *ggpubr* v0.4.0 [121] were used.

Linear mixed effect models fit by restricted maximum likelihood (REML) estimates were used to test hypotheses (consistent with the approach in [61]). Models took the following general form:

$$TPE_i = \beta_0 + \beta_1\ time_i * \beta_2\ adherence_i * \beta_3\ measure + \beta_4\ baseline\ TPE_i + \beta_5\ decline_i + subject_{0i} + \varepsilon,$$

Here, *time* relates to sequential twenty second epochs recorded across the experiment and *adherence* denotes the number of cognitive training sessions completed. *Baseline TPE* was included to account for individual differences in baseline performance (during the first testing

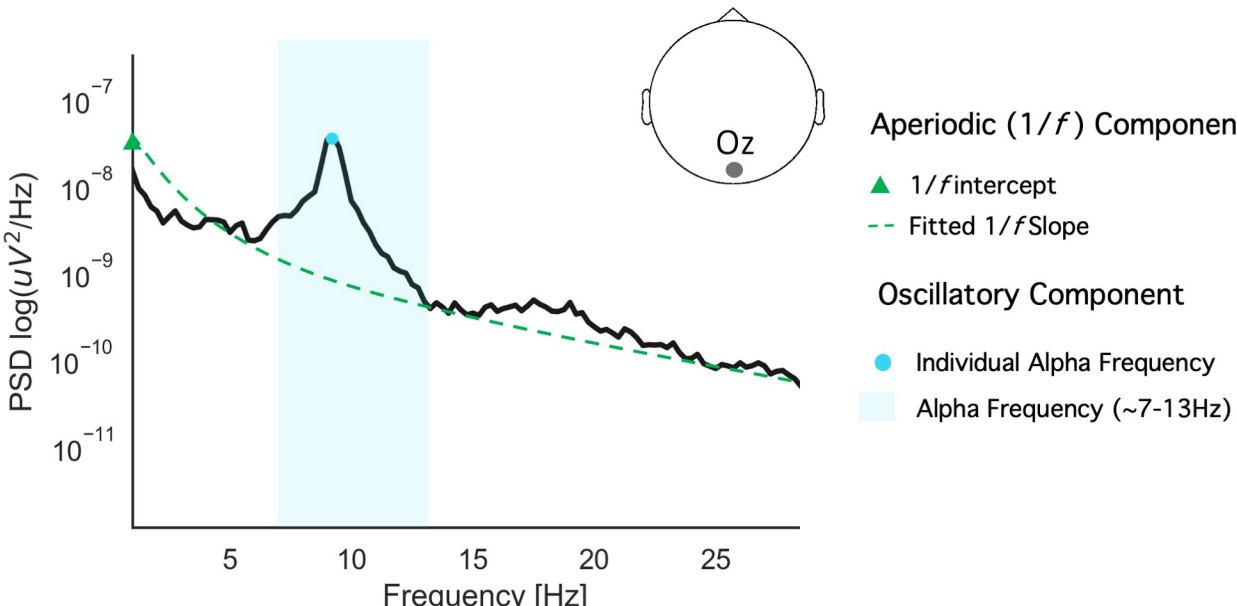

**Fig 2. Example schematic of aperiodic (1/f) and oscillatory activity in the raw EEG trace (mixed signal).** The EEG components of interest are indicated in the legend. Here, the y-axis specifies log-transformed power (i.e., strength of neural activity; higher values indicate greater activity) while frequency (Hz, cycles per second) is represented on the x-axis (values further to the right indicate higher frequency). Note the clear spectral peak in the 10Hz range (representing individual alpha frequency; denoted by a blue circle) and 1/f-like pattern (demonstrated by the green dashed line).

session of the TMA task) and subject was specified as a random effect on the intercept. *Measure* refers to the respective neural or personality measure of interest within each model (e.g., MAAS scores or IAF). *Decline* refers to individual participants slopes in the TPE metric to account for changes in performance within the first 500 seconds (which were highly correlated across all participants; see examples of individual participant data in [61]). Smoothing splines were added to *Time* in each model, to allow for fitted values to capture non-linear trends in the data (see [122] for an overview of the spline function). Asterisks indicate interaction terms, including all subordinate main effects; pluses denote additive terms. *TPE* was specified as the outcome variable. Within our models, categorical factors were sum-to-zero encoded [123], allowing for the directionality of effects to be examined through visualisation [124]. An 83% confidence interval (CI) threshold was used as this corresponds to the 5% significance level with non-overlapping estimates [125,126]. P-value estimates were calculated using Type II Wald $\chi^2$-tests from the *car* package [116]. Simple linear regressions were also used to determine how changes in individual neural metrics related to adherence to cognitive training.

## Results

On average, data recorded on *Google Analytics* demonstrated that participants completed a total of 9.8 mindfulness sessions across the one-week period (SD = 6.98, range = 0–21), with a mean number of 1.47 session per day. A linear regression demonstrated that the average number of sessions completed across all participants increased by 0.045 each day. Overall, participants had lower TPE scores (thus superior performance) in the post-meditation ($M_{TPE}$ = 6987, SD = 2450, range = 2171–11532) than the pre-meditation session ($M_{TPE}$ = 5515, SD = 2383, range = 2427–11287; see Fig 4B). While, on average, participants performed better in their second (post-meditation) session, individual participant data demonstrated that practice effects were not consistent, and some participants performed more poorly in the second than the first session (see individual data points; Fig 4A). For full descriptive statistics of all individual-level variables of interest, see Table 1. Internal consistency of MAAS responses within the current sample was $\alpha$ = 0.89. For correlations between all experimental variables of interest, see Fig 3.

### Resting-state neural metrics

Fig 4 demonstrates changes in neural metrics (IAF and 1/*f* slope and intercept) between the first (pre-meditation) and second (post-meditation) sessions. Fig 5 shows the average neural metrics for each session, by topography. We note that these distributions are similar to comparative samples observed within the literature [61,69,128,129]. For all neural metrics, mean values were consistent (within two decimal points on average) between testing sessions. In the first session, the average IAF was 9.93Hz (SD = 0.67, range = 8.75–11.5), compared to the second session, where the average recorded IAF was 9.94 Hz (SD = 0.58, range = 8.75–11). The mean 1/*f* slope for the first session was -1.02 (SD = 0.34, range = -1.75–-0.22) while for the second session, the average slope was -1.05 (SD = 0.3, range = -1.57 –-0.38). For the 1/*f* intercept,

**Table 1. Means, standard deviations and range of individual-level variables of interest.**

|  | Mean (SD) | Min | Max |
|---|---|---|---|
| Mindfulness Attention Awareness Scale | 3.44 (0.83) | 1.75 | 5.42 |
| Conscientiousness | 29.86 (5.45) | 12 | 40 |
| Neuroticism | 19.70 (3.64) | 12 | 27 |
| Enjoyment | 71.53 (17.67) | 29.14 | 100 |

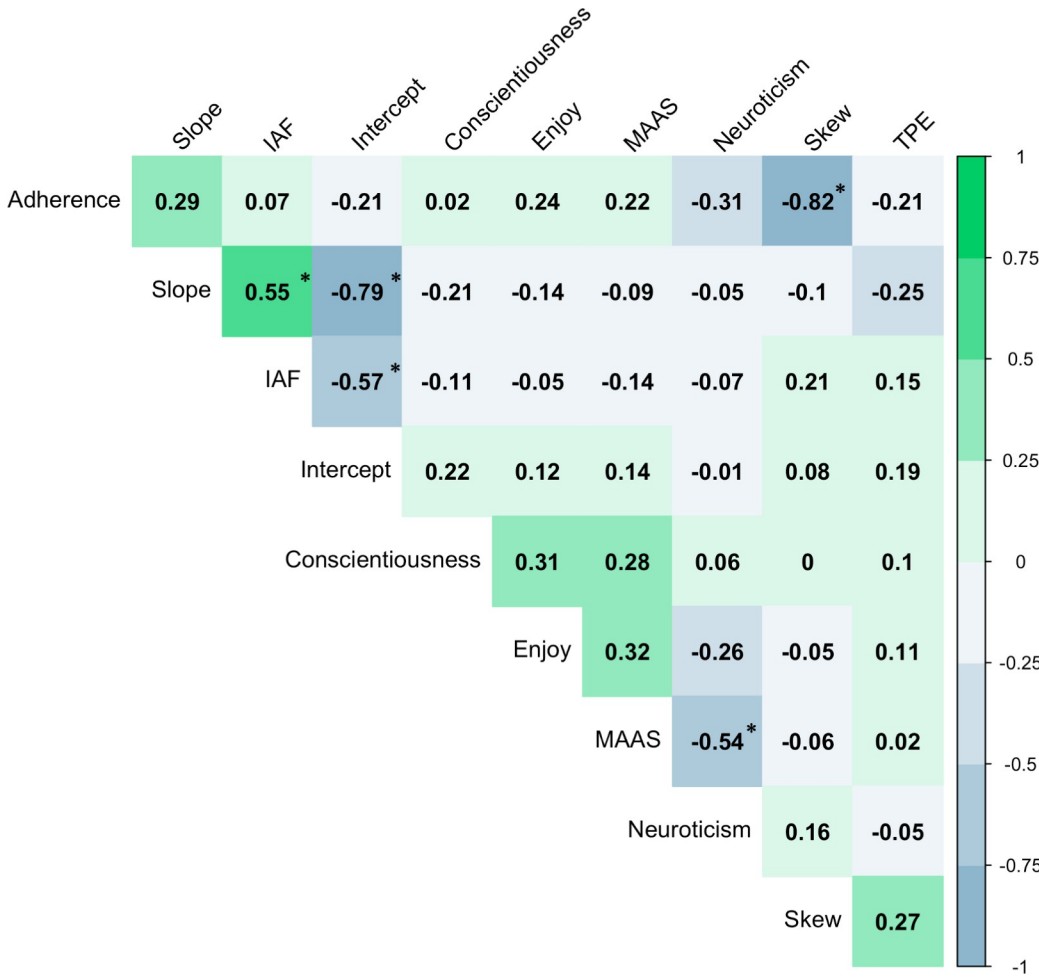

**Fig 3. Relationship between cognitive, personality and neural metrics of interest.** Here, IAF denotes individual alpha frequency, MAAS indicates Mindfulness Attention Awareness Scale scores and TPE is tactical picture error (i.e., performance). Adherence denotes adherence to the cognitive training intervention. Correlogram illustrates the correlation coefficients between each of the metrics of interest. Negative (blue) values indicate a negative association, while positive (green) values indicate a positive association. Values denote Pearson's *r* coefficients. Note that statistically significant associations are denoted with a * ($p < 0.001$). Plot created using the *corrplot* package [127].

the average recorded value in the first session was -25.11 (SD = 0.81, range = -26.55–-23.56) and -25 for the second session (SD = 0.78, range = -26.51–-23.22).

For aperiodic 1/*f* activity, linear regressions demonstrated that levels of adherence to cognitive training did not predict individual resting-state metrics in the second session, when controlling for baseline metrics gathered from the first session. The regression predicting post-1/*f* slope from initial 1/*f* slope and levels of adherence to training was statistically significant ($R^2 = 0.53$, $F(37) = 21.17$, $p < 0.001$). However, the only significant predictor of the second 1/*f* slope was initial 1/*f* slope ($\beta = 0.65$, $p < 0.001$). This was also the case for the 1/*f* intercept ($R^2 = 0.58$, $F(37) = 25.61$, $p < 0.001$) with initial intercept significantly predicting post-intervention 1/*f* intercept ($\beta = 0.75$, $p < 0.001$). For IAF, a significant linear regression ($R^2 = 0.82$, $F(37) = 83.56$, $p < 0.001$) demonstrated that adherence to cognitive training was predictive of second session IAF ($\beta = -0.016$, $p < 0.05$), alongside initial IAF ($\beta = 0.77$, $p < 0.001$). We note that this was only a small coefficient (see model outputs in S2 File).

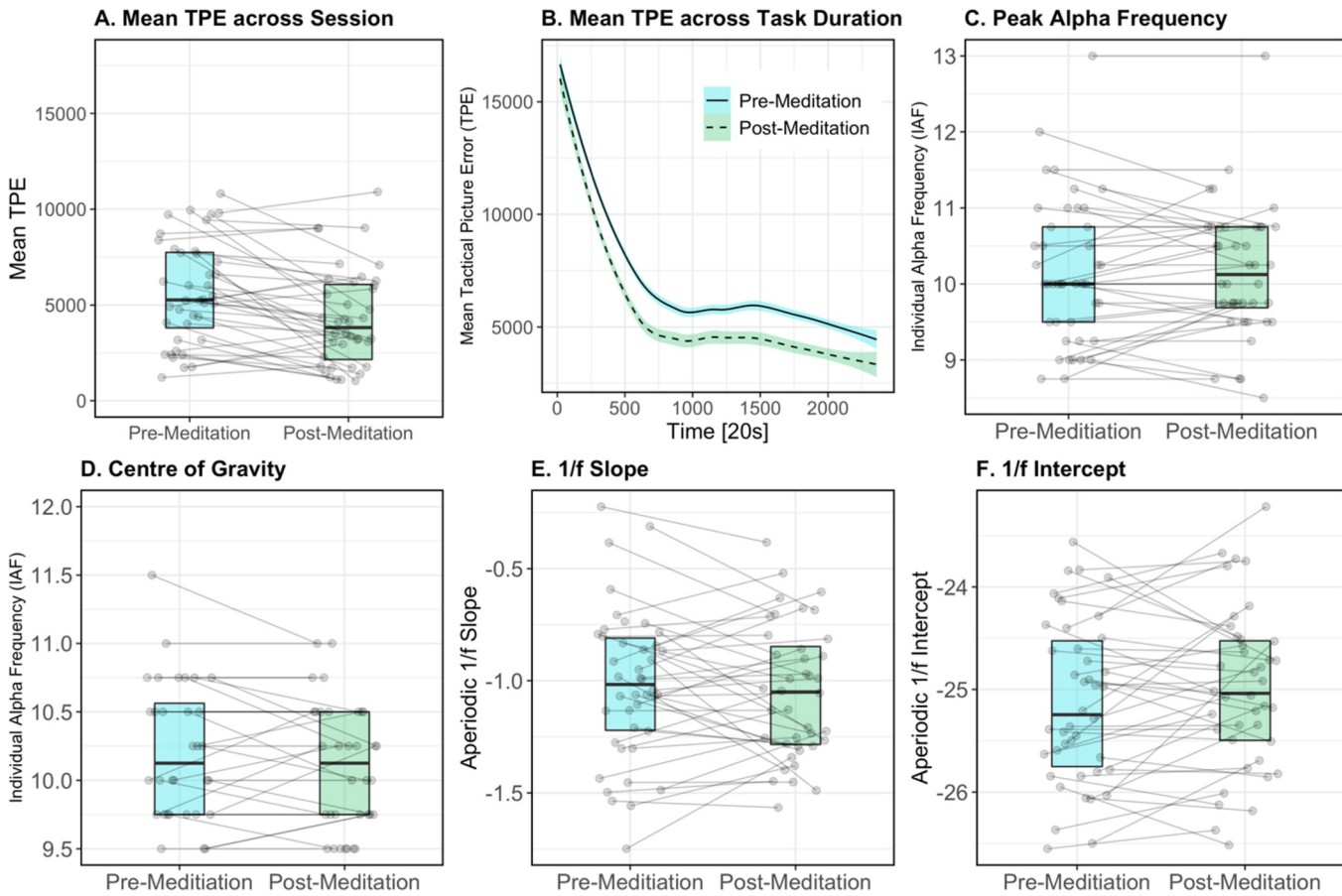

**Fig 4.** Boxplots of mean metrics for tactical picture error (A), individual alpha frequency (peak alpha frequency; (C) and centre of gravity; D), 1/*f* slope (E) and 1/*f* intercept (F). Here, metrics measured in the pre-meditation session (1) are shown in blue, and metrics measured post-meditation intervention are shown in green. Individual data points represent individual participant values. Horizontal lines join participant values from the first (pre-meditation) to second (post-meditation) sessions. Plot (B) demonstrates the mean tactical picture error across experimental time, collapsed across all participants.

### Modelling adherence to cognitive training to predict performance

The first model aimed to determine whether adherence to cognitive training was associated with performance in the post-intervention testing session. Here, we found that TPE in the second session was modulated by the cubic Time term and cognitive training ($\chi^2 (4) = 121.33$, $p < .001$), where greater adherence to cognitive training was related to lower TPE (better performance) in the second testing session. This effect was more pronounced at later time points (see Fig 6A). All model outputs are presented in S3 File.

**Timing of intervention (skew analysis).** To assess whether the timing of individual adherence to the mindfulness-based cognitive training program was related to second session TPE scores, we calculated individual skew for each participants' adherence across the seven days. This measure aimed to capture whether individual participant adherence was positively skewed (i.e., they completed a higher density of training sessions closer to the first lab visit) or negatively skewed (i.e., they completed a higher density of training sessions closer to the second lab visit). Skew metrics were calculated while controlling for individual participant means to avoid the interpretability issues with skew values close to zero (i.e., a skew of zero does not distinguish those who have consistent strict adherence versus consistent low adherence). In this case, we used skew measures derived according to the method proposed by Sinha et al.

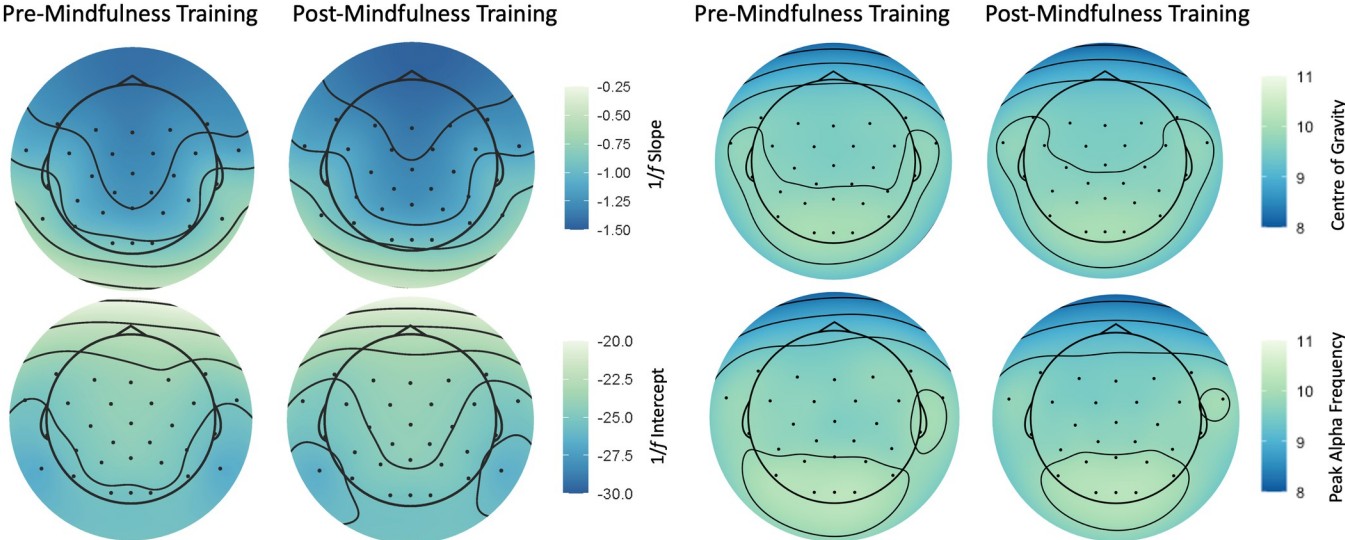

**Fig 5. Topoplots demonstrating average neural metrics across sessions (pre- and post-mindfulness training intervention).** Changes in aperiodic 1/*f* activity (1/*f* slope and intercept) are shown on the left, while changes in individual alpha frequency (both *Peak Alpha Frequency* and *Centre of Gravity*) are shown on the right. Dark blue indicates lower values, while lighter green indicates higher values. Plots created using the *eegUtils* [130] package in R.

[131]. For the current sample, individual participant cognitive training adherence had, on average, a skew of 0.19 (SD = 0.96, range = -1.62–1.62). When modelling skew as a predictor of TPE across time (controlling for baseline performance, decline, and individual participant variation), we observed a significant interaction of Time, cognitive training adherence and skew ($\chi 2$ (4) = 25.25, $p$ < .001). Here, we found that negative skew (late adherence) was related to larger effects of levels of adherence, where greater adherence was related to superior performance. For positive skew (early adherence), levels of adherence did not differentiate performance as noticeably. However, collapsing across total levels of adherence, the model suggests that early adherence was related to superior TPE scores, when compared to late adherence. See Fig 6B for visualisations of the modelled data.

To further analyse whether mindfulness prior to participants' first sleep was influential in predicting outcomes, we also coded individual participants as 1 and 0 (depending on if they engaged in mindfulness prior to first sleep or not) and included this factor into the model. In the current study, there were 22 participants who completed at least one mindfulness session prior to their first sleep (i.e., on the first day of training), while 18 participants did not engage in any sessions until the second day of the intervention. We found that training prior to first sleep did significantly predict outcomes (Time x Adherence x Prior Sleep ($\chi 2$ (4) = 36.57, $p$ < .001), but did not change the overall pattern (where greater adherence was related to superior performance for most time points). Here, results demonstrated that lower levels of adherence to the cognitive training were still related to better performance (when compared to low levels; starting after the first day) if mindfulness exercises were completed on the first day, prior to a participant's first sleep. See Fig 6C for modelled effects.

**Enjoyment.** To assess whether self-reported enjoyment of the mindfulness-based cognitive training exercises influences cognitive training outcomes, we created an average enjoyment score for each participant across all seven days. The model predicting TPE from levels of adherence alongside enjoyment demonstrated a significant interaction of Time, cognitive training adherence and enjoyment ($\chi 2$ (4) = 80.59, $p$ < .001). Here, we observed that higher levels of enjoyment were related to more distinguished effects of levels of adherence. At lower

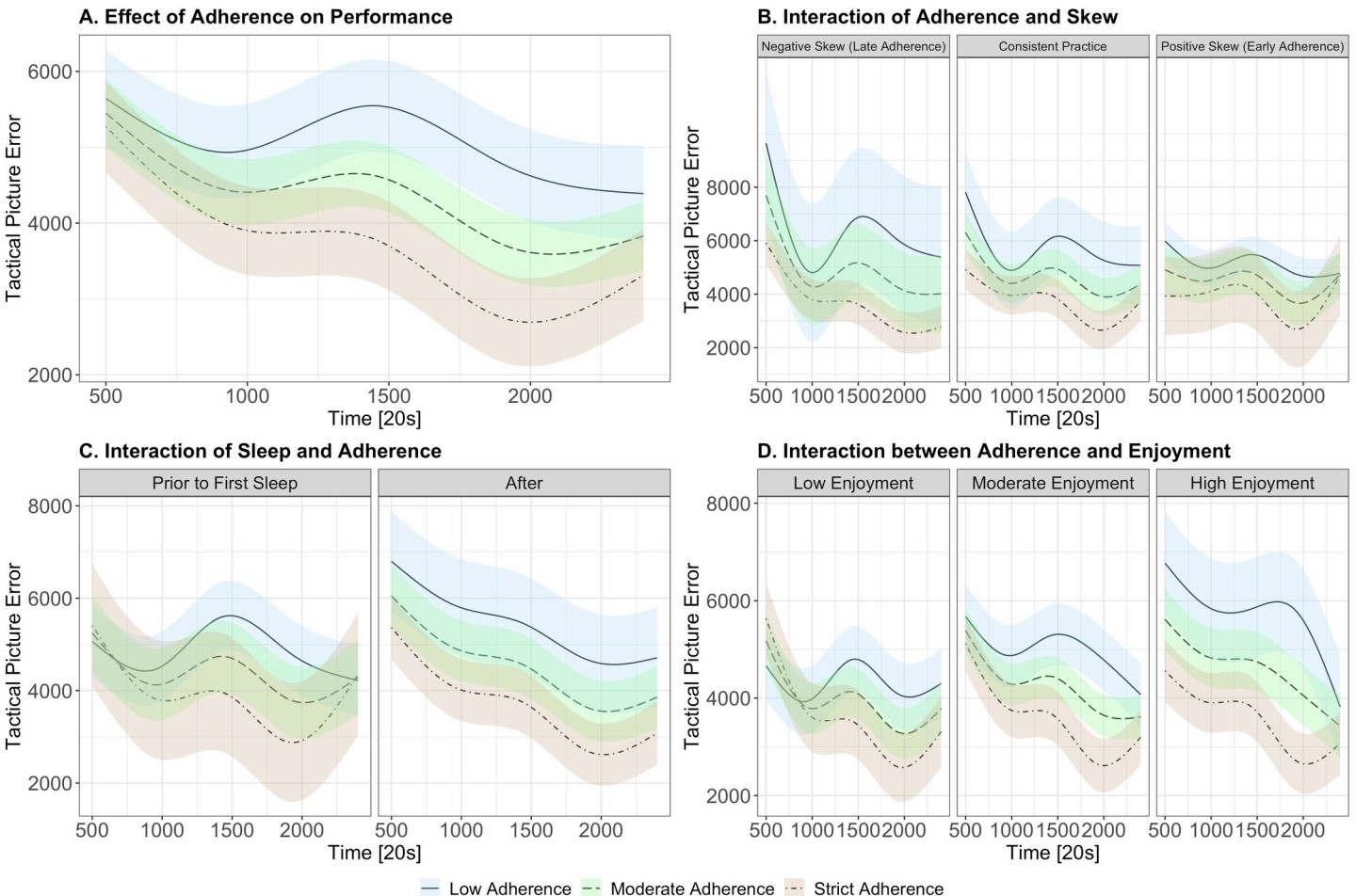

**Fig 6.** Modelled effects of (A) cognitive training adherence, (B) timing of adherence (skew), (C) prior sleep and (D) enjoyment on TPE. TPE is represented on the y-axis (higher scores indicate poorer performance), while experimental time is represented on the x-axis. Colours indicate varying levels of adherence to the cognitive training intervention (beige indicates high levels of adherence, green for moderate and blue for low adherence). Facets represent varying levels of variables of interest. Differences in factors are categorised as low, moderate and high based on the sample's first quartile, median and third quartile, respectively. Statistical models included these predictors as continuous variables, however, categories were created for visualisation purposes. For comparative purposes, we note that as a general 'practice effect' (i.e., model predicted values when adherence = 0), participants demonstrate an average TPE of 5541 across experimental time.

levels of enjoyment, data suggested that individuals demonstrated superior performance (when collapsing across levels of adherence), when compared to greater levels of enjoyment. See Fig 6D for modelled effects.

**MAAS scores.** To investigate whether levels of dispositional mindfulness modulated the effects of cognitive training, we modelled TPE against MAAS scores. Modelling revealed a significant interaction between Time, MAAS scores and adherence to cognitive training ($\chi 2$ (4) = 99.06, $p < .001$), whereby the differential effects of cognitive training were more pronounced in lower/moderate levels of dispositional mindfulness (i.e., greater adherence was associated with better performance), but higher MAAS scores were related to lower TPE scores when collapsed across adherence. See Fig 7A for modelled effects.

**Personality traits.** We modelled the effects of individual levels of conscientiousness and neuroticism to determine how personality may interact with benefits gained from mindfulness-based cognitive training interventions. Here, we observed a significant interaction of Time, cognitive training adherence, neuroticism and conscientiousness ($\chi 2$ (4) = 131.73,

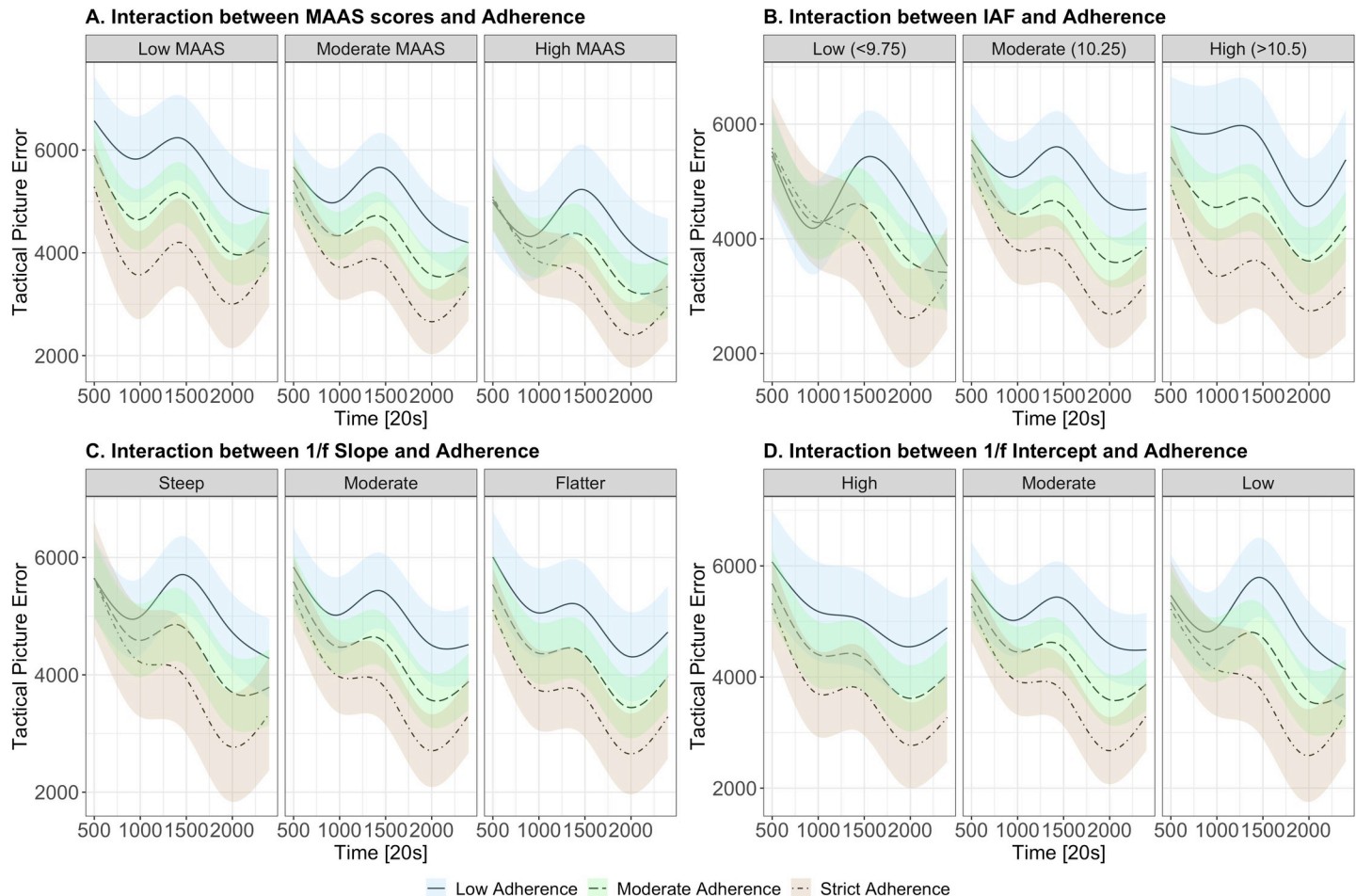

**Fig 7.** Modelled effects and interactions of cognitive training adherence with (A) mindfulness attention awareness scores, (B) individual alpha frequency, (C) 1/$f$ slope and (D) 1/$f$ intercept. TPE is represented on the y-axis (higher scores indicate worse performance), while experimental time is represented on the x-axis. Colours indicate varying levels of adherence to the cognitive training intervention (beige indicates high levels of adherence, green for moderate and blue for low adherence). Facets represent varying levels of neural metrics. Differences in neural metrics are categorised as low, medium, and high or steep, moderate and flatter based on the first quartile, median and third quartile, respectively. Statistical models included these predictors as continuous variables, however, categories were created for visualisation purposes.

$p < .001$). While a significant effect was reported, visualisations demonstrated only very subtle effects of these variables, with the overarching pattern (greater adherence associated with superior scores) still evident across all levels of personality metrics. We note that this model provided the best model fit (see Table 2 for model fit comparison), however all models demonstrated $R^2$ values of $>.69$ indicating good fit overall. See S1 Fig for modelled effects of personality traits.

Overall, these findings suggest that the most prevalent factor predicting second session performance was the level of adherence (i.e., number of sessions). While models demonstrated small nuances of individual factors (MAAS, personality, enjoyment), higher adherence was consistently related to better performance. Notably, our models demonstrated that timing of adherence did exert influence on TPE scores, with early adherence, especially prior to a participant's first sleep, associated with greater performance outcomes irrespective of total volume of training.

**Neural metrics.** Next, we investigated whether resting-state neural metrics interact with the effect of cognitive training. The model of IAF demonstrated a significant interaction of the

**Table 2. Comparison of all model fits.**

| Model | R² | R² (marginal) | ICC | RMSE | Sigma | AIC weights | BIC weights |
|---|---|---|---|---|---|---|---|
| Personality* | 0.752 | 0.289 | 0.651 | 1513.674 | 1528.482 | < 0.001 | < 0.001 |
| Skew | 0.695 | 0.273 | 0.58 | 1638.875 | 1651.59 | 1 | 1 |
| IAF | 0.728 | 0.309 | 0.606 | 1598.025 | 1610.031 | < 0.001 | < 0.001 |
| MAAS | 0.726 | 0.313 | 0.602 | 1615.519 | 1627.852 | < 0.001 | < 0.001 |
| Prior Sleep | 0.724 | 0.295 | 0.609 | 1608.571 | 1620.656 | < 0.001 | < 0.001 |
| Intercept | 0.723 | 0.289 | 0.611 | 1611.679 | 1623.788 | < 0.001 | < 0.001 |
| Adherence** | 0.709 | 0.29 | 0.59 | 1633.233 | 1643.75 | < 0.001 | < 0.001 |

*Note*. All models follow a similar structure ($TPE_i = \beta_0 + \beta_1\ time_i * \beta_2\ engagement_i * \beta_3\ measure + \beta_4\ baseline\ TPE_i + \beta_5\ decline_{0i} + \varepsilon,$). Models are ordered according to best fit, however, note that not all models were fit to identical datasets (i.e., missing values may be different). Model's information criterion (IC) weights are calculated as per the *performance* package in R [117], whereby IC weights are computed as exp(-0.5*delta_ic)/sum(exp(-0.5*delta_ic). Here, *delta_ic* denotes the difference between the model's IC value and the minimal IC value of all comparative models [132]. *Personality includes conscientiousness and neuroticism as interacting factors. **Adherence model includes no other individual factors.

cubic Time term, initial IAF and adherence to cognitive training ($\chi 2\ (4) = 137.74$, $p < .001$). Across all levels of IAF, there was a consistent pattern of improved performance with higher levels of adherence to the cognitive training regime. For lower levels of IAF, however, performance was more similar across levels of adherence, when compared to moderate and high, particularly within the first half of the testing session (see Fig 7B). Additionally, collapsing across levels of adherence, performance for those with lower IAF appears to be better overall when compared to those with higher IAF.

For the 1/*f* slope, we found a significant interaction of Time, pre-intervention 1/*f* slope and adherence to cognitive training on performance in the TMA task ($\chi 2\ (4) = 50.26$, $p < .001$). These results showed that greater adherence to cognitive training was consistently associated with greater performance across all levels of the 1/*f* slope. However, these differences were more pronounced for flatter 1/*f* slopes, in comparison to steep and moderate slopes (see Fig 7C).

Lastly, we modelled the relationship between 1/*f* intercept and performance on the TMA task, noting a significant interaction between Time, pre-intervention 1/*f* intercept and adherence to cognitive training ($\chi 2\ (4) = 60.71$, $p < .001$). Here, we observed that superior performance related to adherence to the cognitive training regime was most distinguishable in higher 1/*f* intercepts. However, for lower intercepts, performance appeared to converge most consistently across time.

Overall, our findings indicated that individual levels of resting-state neural metrics did not interact with the effects of mindfulness-based training (whereby, for most, stricter adherence was consistently related to better performance). These results highlight the potential generalisability of mindfulness-based training. However, we note there were subtleties in the magnitude of this effect, with some levels of neural metrics (lower IAF and higher 1/*f* intercept) related to less distinguishable performance outcomes associated with less adherence. For these groups, performance appeared to be more similar across levels of adherence at the beginning of the testing period, but separated at later time points. This may demonstrate that for these individuals, enhancements from mindfulness training are more prominent at longer durations of task participation.

## Discussion

The potential for mindfulness training to enhance cognitive function has drawn increased interest for a range of applications. However, two key issues are present within the

mindfulness-based cognitive enhancement literature. Firstly, mixed findings have been summarised in recent systematic reviews [8,33] and discussions are highlighting how an individual differences approach is crucial to reconcile these inconsistencies [19]. Second, studies are typically limited to traditional cognitive tests, and do not demonstrate transferability to more complex and dynamic situations. The current study aimed to investigate how a one-week mindfulness-based cognitive training regime can augment performance in more dynamic tasks (i.e., the target motion analyst task [23]) while recognising the influence of individual differences. Using linear-mixed effect models with cubic splines, we showed that greater adherence (i.e., higher number of sessions) to the mindfulness-based exercises was associated with superior performance in the second testing session, when compared to lower levels of adherence. Further, our findings demonstrated that this effect was consistent across all levels of resting-state EEG metrics, and was not accompanied by individual changes in these metrics across time. In line with our hypotheses, some individual factors were significantly associated with the results, however, the most consistent predictor among models was adherence to the cognitive training regime.

## Adherence to cognitive training

Our study employed a one-week mindfulness-based training regime to determine how individual levels of adherence may relate to enhancements in cognitive performance. Here, participants self-selected the number of times they engaged in the guided mindfulness practices, from 0 to 21 sessions of the prescribed 5-minute exercises. This opposes the commonly employed between-groups approach in mindfulness-based cognitive training research, in the endeavour to better characterise the effect of volume of training. Our results showed that greater adherence to the cognitive training regime was associated with superior performance in the TMA task (when accounting for individual baseline performance) and this effect was consistent even across multiple additions of interacting factors (discussion below). These results extend on the multiple studies examining the effects of mindfulness-based training on traditional laboratory tasks (e.g., [10,17,24–26]) and provide evidence for the transferability of this type of training even to dynamically changing settings. These findings speak to a 'more is better' but 'some is always good' approach to mindfulness engagement, with performance enhancements directly proportionate to all levels of adherence. Such results could also be considered in relation to findings from studies employing single-session mindfulness practices prior task engagement [31,36,133]; raising questions about the differences in how performance benefits are derived from brief mindfulness practice prior to task engagement, versus higher volumes of more sustained practice over time.

Strikingly, our training regime was correlated with individual enhancements in performance even across a short time frame (up to 15 minutes per day, for one week), consistent with Zeidan et al. [24]. In our study, participants were able to undertake exercises on their own accord, and our use of *Google Analytics* helped to minimise bias of self-reporting of adherence to the intervention. While the use of online audio recordings meant that we were unable to account for individual participant surrounds, our approach provided a practical and accessible way for individuals to engage in cognitive training. From this perspective, our findings highlight how mindfulness-based cognitive training may be a useful and accessible way to improve performance across a variety of cognitively demanding contexts.

## Timing of intervention adherence

To further characterise the neural correlates of mindfulness-based cognitive training, we examined how the timing of adherence might influence its effects. From this perspective, we

explored the timing of an individual's density of mindfulness exercise (i.e., whether more training was completed near the first testing session versus nearer the second testing session). Secondly, as an exploratory analysis, we tested differences in how mindfulness practice completed on the first day of training (after learning the target motion analyst task; and before sleep), versus practice beginning on the second day, influenced the observed relationships. For the timing of adherence (measured by skew of participants' adherence), we found the same pattern of higher levels of adherence (more = better) in individuals who were late adherers (that is, those with a higher density of sessions were completed closer to the second testing session). For those who were early adherers, performance was overall better across all levels of adherence, when compared to later adherers. This finding was consistent with the results of our first day/prior to sleep analysis, where effects of adherence were more pronounced if participants did not complete a mindfulness session on the first day; and those who completed an exercise on the same day of learning the task, performed better overall (regardless of total level of adherence). These findings allow us to theorise potential mechanisms of mindfulness engagement and appear to speak to a strong consolidation-based effect of the training [18,98]. That is, mindfulness training can enhance performance if completed shortly after learning a new task, irrespective of the volume of training across the total intervention period. However, volume of training might still be important when individuals miss this crucial early window, with higher volumes of training still augmenting performance, perhaps by creating optimal attention states for performance at the time of testing [36,134].

### Enjoyment

In addition to timing of adherence, we investigated how individual-level enjoyment of the mindfulness exercises was associated with performance outcomes. As suggested by Tang and Braver [19], individual preferences for certain types of mindfulness practices can influence adherence. In our sample, however, we did not observe a relationship between self-reported enjoyment and adherence to the cognitive training regime. Our analysis did demonstrate that enjoyment was a significant predictor of performance; but again, interacting effects of enjoyment only altered the magnitude of variation between low and higher levels of adherence, and did not change the overall main effect of adherence. We do acknowledge, however, that individual enjoyment reports may be influenced by social desirability bias. Nevertheless, it is hoped that these effects are mitigated by the use of online reporting tools, in which participants are not required to interact with the researcher.

### Dispositional mindfulness

Next, we examined how dispositional mindfulness (as measured through the Mindful Attention Awareness Scale [39]) interacted with cognitive training to predict performance. Consistent amongst results in other psychological interventions [37,38,135], we found that individual levels of dispositional mindfulness were related to enhancements associated with the mindfulness-based training program. Here, we found that lower levels of dispositional mindfulness were correlated with more distinct differences in lower and higher levels of adherence (similar to higher IAF, flatter slopes and higher $1/f$ intercepts; see below). That is, across the entire testing session, greater adherence was related to superior performance. For individuals with higher self-reported dispositional mindfulness, the effect of adherence was pronounced only towards the second half of the testing session, with more similar performance at the beginning of testing. However, when disregarding the main effect of adherence, those with higher self-rated trait mindfulness were likely to demonstrate improved performance overall, when compared to those with lower trait mindfulness. These results are consistent with findings of

Anicha et al. [136], Ruocco and Wonders [137] and Schmertz et al. [138], where self-reported trait mindfulness has been shown to relate to cognitive function (i.e., less omissions or better sustained attention during testing). Our results provide converging support for Tang and Braver's [19] proposal that those with high trait mindfulness may already be at the upper capacity of their attentional capacity, whereby further mindfulness-based training does not result in further improvement to their abilities. Nevertheless, while those with high MAAS had converging performance at the beginning of the testing session, fluctuations in performance across time demonstrated that greater adherence was still related to better performance in the second half of the testing session, suggesting that adherence to the cognitive training regime still enhanced sustained attentional capacity to some extent.

## Individual personality traits

As previous studies have further demonstrated relationships between personality traits and intervention outcomes, we examined how conscientiousness and neuroticism were related to performance enhancements. Unlike Anderson [47], Chapman et al. [48] and Sanderson and Clarkin [49], we found that these personality traits were not predictive of individual adherence to the intervention. However, consistent with other studies [44–46], we were able to replicate the negative association between neuroticism and dispositional mindfulness scores. This finding is congruent with the relationship between neuroticism and poorer subjective well-being [139–141], while mindfulness is known to mitigate neurotic behaviours [3,142] and is related to enhanced psychological well-being [143]. When examining how these personality factors might interact with the effect of adherence, we did find significant interactions of conscientiousness, neuroticism and adherence on performance. However, we note that these effects were very subtle differences in performance across time, and the main effect of greater adherence relating to better performance was most prominent across all interactions of conscientiousness and neuroticism. Again, these findings highlight how mindfulness-based training regimes may provide cognitive benefits for a wide range of individuals, irrespective of baseline personality traits or individual preferences.

## Individual resting-state EEG metrics

In an effort to illuminate neural mechanisms that may underlie performance enhancements from cognitive training, we explored how individual resting-state EEG metrics (IAF and $1/f$) interacted with mindfulness-based training adherence to predict performance on the TMA task. Here, we found that resting-state neural metrics did not change across sessions; that is, distributions were similar across pre- and post-intervention testing, regardless of individual involvement in the training regime. Our analyses demonstrated that individual resting-state metrics in the second session were only predicted by the same measures at baseline testing, with no effects of adherence. These findings speak to the test re-test stability of IAF [144–146] and $1/f$ activity [147,148] demonstrated in other neuroscientific studies. More specifically, these results align with findings of cognitive training on a larger scale (i.e., 100 hours) by Grandy et al. [144] where IAF was stable across testing sessions. Together, these outcomes demonstrate that EEG resting-state metrics appear to be resistant to acute and longer-term cognitive training interventions. As such, these metrics may serve as viable "fingerprinting" tools, whereby individuals can be reliably identified across time (as discussed in biometric research, e.g., [149–151]).

However, we note that the short-term nature (only one-week) of mindfulness-based training may be the catalyst for these null findings. Previous studies that have observed differences in the EEG of long-term meditators [95,96] have investigated individuals with years of

experience (e.g., mean of 20 years) and analysed recordings *during* meditation. It may be the case that mindfulness practice does not relate to long-term resting-state EEG changes. Alternatively, it could also be that 1.5 hours of mindfulness-based training is not a sufficient catalyst for enduring EEG change. Nevertheless, irrespective of changes in resting-state neural metrics, we discovered that performance enhancements were still evident. These findings highlight the need for future research to explore other avenues of neurophysiological change that may be underlying benefits from this style of cognitive training. For example, individuals may exhibit variations in oscillatory activity in-task (relating to attention or inhibition [152]. Alternatively, other physiological changes from mindfulness, like heart rate variability [153,154]—also related to cognition [155–157]—may provide insights into the underlying mechanisms of performance improvements.

On top of this, we found that individual neural characteristics at baseline were not associated with the general effects of mindfulness adherence. That is, all individual levels of IAF and $1/f$ activity were still related to benefits from stricter adherence to the cognitive training regime. There were, however, small subtleties demonstrated across different levels of individual resting-state neural metrics. For example, higher levels of IAF, flatter slopes and higher $1/f$ intercepts demonstrated more distinct relationships to levels of high versus low adherence (i.e., more adherence was correlated with superior performance and less adherence was correlated with poorer performance across the entire testing session). Contrastingly, for steeper $1/f$ slopes, lower IAF and lower $1/f$ intercepts, performance was more likely to converge at certain time points, even though similar trends were present overall. Nevertheless, this shows promise for mindfulness-based training programs to be widely beneficial across different 'types of brains'.

## Limitations and future directions

While our study has demonstrated a positive effect of mindfulness-based cognitive training interventions, we note a few limitations to its rigour. Previous meta-analyses and reviews in the mindfulness literature have suggested that many studies fail to define meditation styles, creating difficulties in the comparison of different techniques [158,159]. Within our study, we did not implement one specific type of mindfulness (e.g., focused attention, open-monitoring or loving-kindness) but used a variety of mindfulness techniques across the week. From this perspective, we are unable to identify which particular mental processes within mindfulness practice help to enhance performance. Similarly, while the study highlights the intricate relationship between mindfulness interventions and cognitive enhancement, we did not directly examine whether participants showed higher levels of dispositional mindfulness following the intervention. Future approaches should aim to thoroughly examine mindfulness' differential impacts on cognition, trait mindfulness, and subsequent inter-relationships. We note that, overall, this study aimed to be more exploratory in nature, in that we investigated outcomes in more dynamic and complex settings. However, we look to future research to elucidate in more detail the ways in which distinct mindfulness practices work and to better understand their benefits to cognition more longitudinally.

Moreover, while we briefly examine how participant enjoyment may play a role in the benefits gained from the mindfulness practice, we did not account for individual preferences for different modalities. Anderson and Farb's [135] experimental research has demonstrated that individual differences in preference of modality of mindfulness interventions can help to reduce attrition, facilitate adherence and enhance outcomes (similar to findings in clinical practices too [160]). Thus, a tailored approach to individual participant training programs could help to increase levels of engagement. Supplementary to this, our study could be biased by the self-selection of participants to engage in the intervention, with largely fluctuating levels

of adherence. However, this did fit well with our analytical approach, where we used levels of adherence as a continuous predictor. Further, as higher levels of adherence and greater performance outcomes were not related to enjoyment of the practice, we can suggest that it is more than a desire or liking for the practice that drives enhancements.

Lastly, while the study aimed to investigate the transferability of mindfulness-based cognitive training exercise, we do note the extremely novel and dynamic nature of the target motion analyst task [23] used within the experiment. Here, it is difficult to explicitly quantify the difficulty of task demands, and thus a comprehensive task analysis could help to better identify how mindfulness-based training techniques help to improve dynamic performance. However, we hope that future research can help to provide insight into the gaps where mindfulness-based training enhancements diverge from simple/traditional cognitive tasks to undertakings of larger complexity.

## Conclusion

The current study examined how a short-term mindfulness-based cognitive training regime augments performance on a dynamic cognitive task. Our findings demonstrated that stricter adherence to the mindfulness-based training regime was associated with superior performance in the second testing session, even when controlling for baseline performance and individual differences. In addition to this, we explored how individual moderators (such as enjoyment, timing of adherence and personality) influenced the effects of level of adherence. Our analyses demonstrated that resting-state EEG was not a predictor of performance outcomes. Further, while subtle effects of individual factors were evident, adherence remained the most prominent predictor of second session performance. We also noted that the pattern of training adherence contributed to performance augmentation; adherence to early mindfulness training sessions was associated with heightened task performance overall. Nevertheless, higher volumes of cognitive training at later time points were still related to enhanced performance. In summary, the present findings illustrate the propensity of mindfulness-based training to transfer cognitive enhancements to performance on an unrelated complex and dynamic task. Practically, the present work has demonstrated an effective solution to implement an online mindfulness-based training program through a web browser approach, affording objective monitoring of individual training adherence.

## Supporting information

**S1 Fig. Modelled effects of conscientiousness, neuroticism and levels of adherence on second session TPE scores.** Modelled effects of conscientiousness and neuroticism on TPE scores. TPE is represented on the y-axis (higher scores indicate poorer performance), while experimental time is represented on the x-axis. Colours indicate varying levels of adherence in the cognitive training intervention (brown indicates high levels of adherence, green for moderate and blue for low adherence). Facets represent varying levels of neuroticism (left to right = low to high) and conscientiousness (top to bottom = low to high). Differences in factors are categorised as low, moderate and high based on the sample's first quartile, median and third quartile, respectively. Statistical models included these predictors as continuous variables, however, categories were created for visualisation purposes.
(PNG)

**S1 File. CRUSE practice session speech.**
(DOCX)

**S2 File. Regression model outputs.**
(DOCX)

**S3 File. Linear mixed effect model outputs.**
(DOCX)

## Author Contributions

**Conceptualization:** Chloe A. Dziego, Ina Bornkessel-Schlesewsky, Matthias Schlesewsky, Ruchi Sinha, Maarten A. Immink, Zachariah R. Cross.

**Data curation:** Chloe A. Dziego.

**Formal analysis:** Chloe A. Dziego.

**Funding acquisition:** Ina Bornkessel-Schlesewsky, Matthias Schlesewsky, Ruchi Sinha, Maarten A. Immink.

**Methodology:** Chloe A. Dziego, Ina Bornkessel-Schlesewsky, Matthias Schlesewsky, Maarten A. Immink.

**Resources:** Maarten A. Immink.

**Software:** Chloe A. Dziego.

**Supervision:** Ina Bornkessel-Schlesewsky, Matthias Schlesewsky, Ruchi Sinha, Maarten A. Immink, Zachariah R. Cross.

**Visualization:** Chloe A. Dziego.

**Writing – original draft:** Chloe A. Dziego.

**Writing – review & editing:** Chloe A. Dziego, Ina Bornkessel-Schlesewsky, Matthias Schlesewsky, Ruchi Sinha, Maarten A. Immink, Zachariah R. Cross.

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
