## [Decision Letter · Decision Letter 0]

19 Jan 2024

PONE-D-23-29669Augmenting complex and dynamic performance through mindfulness-based cognitive training: an evaluation of training adherence, trait mindfulness, personality and resting-state EEG.PLOS ONE

Dear Dr. Dziego,

Thank you for submitting your manuscript to PLOS ONE. After careful consideration, we feel that it has merit but does not fully meet PLOS ONE’s publication criteria as it currently stands. Therefore, we invite you to submit a revised version of the manuscript that addresses the points raised during the review process.

We look forward to receiving your revised manuscript.

Kind regards,

Thalia Fernandez, Ph.D.

Academic Editor

PLOS ONE

Journal Requirements:

"Preparation of this work was supported by a grant from the Defence Science and Technology Group [Research Agreement 9208] under the Research Network for Undersea Decision Superiority (RN-UDS; https://www.dst.defence.gov.au/partner-with-us/university/rn-uds#:~:text=DSTG%20is%20working%20with%20Australian,decision%2Dmaking%20on%20Australian%20submarines.). The funders had no role in study design, data collection and analysis, decision to publish, or preparation of the manuscript. IB-S is supported by an Australian Research Council Future Fellowship (FT160100437)."

3. In the online submission form, you indicated that your data will be submitted to a repository upon acceptance.  We strongly recommend all authors deposit their data before acceptance, as the process can be lengthy and hold up publication timelines. Please note that, though access restrictions are acceptable now, your entire minimal  dataset will need to be made freely accessible if your manuscript is accepted for publication. This policy applies to all data except where public deposition would breach compliance with the protocol approved by your research ethics board. If you are unable to adhere to our open data policy, please kindly revise your statement to explain your reasoning and we will seek the editor's input on an exemption. 

4. We note that Figure 1 in your submission contain copyrighted images. All PLOS content is published under the Creative Commons Attribution License (CC BY 4.0), which means that the manuscript, images, and Supporting Information files will be freely available online, and any third party is permitted to access, download, copy, distribute, and use these materials in any way, even commercially, with proper attribution. For more information, see our copyright guidelines: http://journals.plos.org/plosone/s/licenses-and-copyright.

Additional Editor Comments:

The reviewers and I agree that the authors address an important research question about the effect of mindfulness on cognitive outcomes. A strength of the study, in order to make the results more ecological, was to use a complex and dynamic cognitive task

However, I have several concerns regarding the EEG.

1. Since the EEG is being explored in a resting condition, I consider it advisable to compare the EEG of each subject with the electroencephalographic norms to determine if they are within normal limits. Inclusion of participants with abnormal EEG features, often associated with neurological or psychiatric disorders, could severely bias the results.

2. In the Introduction it should be very clear, on the one hand, that the different recording conditions (rest, meditation, different types and difficulties of cognitive tasks) have characteristic EEG patterns that are different from each other. On the other hand, since from the resting EEG several study variables are derived, the reader should be informed what an increase (or decrease) in theta or alpha activity would represent in terms of a healthier state of intrinsic brain functioning. brain (e.g. For example, in the resting EEG is it desirable that theta power increases with the intervention?)

If it had not been previously demonstrated that this task has no learning effect, this question should be addressed within the limitations of the study, i.e., to what extent certain tasks have a learning effect and, therefore, performance will be better in the 2nd. application. In this case, it is desirable that the authors make some recommendations for future studies.

HYPOTHESIS: According to what was mentioned in the Introduction regarding [60, 64], it seems contradictory to expect performance to improve on more complex and dynamic tasks and, simultaneously, to expect the negative slope of 1/f to increase.

Reviewers' comments:

Reviewer's Responses to Questions

**Comments to the Author**

1. Is the manuscript technically sound, and do the data support the conclusions?

Reviewer #1: Yes

Reviewer #2: No

2. Has the statistical analysis been performed appropriately and rigorously? 

Reviewer #1: Yes

Reviewer #2: I Don't Know

3. Have the authors made all data underlying the findings in their manuscript fully available?

Reviewer #1: Yes

Reviewer #2: Yes

4. Is the manuscript presented in an intelligible fashion and written in standard English?

Reviewer #1: Yes

Reviewer #2: Yes

5. Review Comments to the Author

Reviewer #1: Major Strengths:

Innovative Approach: The study's integration of mindfulness-based cognitive training and its impact on cognitive performance in complex and dynamic settings is highly innovative. The use of a short-term intervention to observe significant outcomes is noteworthy.

Comprehensive Data Analysis: The use of linear mixed-effect models with cubic splines to analyze data, considering various factors such as individual EEG metrics and personality traits, demonstrates a thorough analytical approach.

Robust Experimental Design: The study's design, which includes pre- and post-intervention testing and the use of a control task (TMA task), is well-structured and allows for a clear examination of the effects of mindfulness training.

Detailed Consideration of Individual Differences: The investigation of individual differences in mindfulness, personality traits, and EEG metrics is commendable and adds depth to the study's findings.

Areas for Improvement:

Clarification on Mindfulness Training Regime: The manuscript could benefit from more detailed information on the specific mindfulness techniques used, considering the variety of approaches within mindfulness practice.

Expansion on EEG Metric Analysis: While the manuscript addresses resting-state EEG metrics, further exploration into task-related EEG changes due to mindfulness training could provide additional insights.

Consideration of Long-Term Effects: The study focuses on short-term interventions. Including a follow-up assessment could provide valuable information on the longevity of the cognitive benefits observed.

Inclusion of a Control Group: The study could be strengthened by including a control group that does not receive mindfulness training to better attribute observed changes specifically to the mindfulness intervention.

Broader Participant Demographics: The participant age range is relatively narrow (18 - 38 years). Expanding this range could enhance the generalizability of the findings.

Methodological Considerations:

Addressing Potential Bias in Self-Reported Data: The study relies on self-reported adherence and enjoyment of the mindfulness practice. Consideration of potential biases in these reports should be discussed.

Overall Assessment:

This manuscript offers valuable insights into the potential of mindfulness-based cognitive training to enhance performance in complex cognitive tasks. The comprehensive analysis and consideration of individual differences are particularly commendable. However, the study could benefit from further methodological details, consideration of long-term effects, and inclusion of broader participant demographics.

Reviewer #2: Augmenting complex and dynamic performance through mindfulness-based cognitive training: an evaluation of training adherence, trait mindfulness, personality and resting- state EEG.

The authors address an important research question regarding the effect of mindfulness on cognitive outcomes. The main purpose of this study was to examine how a week-long mindfulness intervention influenced the outcome of cognitive performance during the TMA task, while controlling for personality and mindfulness traits. A complex and dynamic cognitive task was measured as the outcome for better generalizing to real world cognitive behaviors.

There are some major questions for this study, as well as the interpretation of the findings. Specifically, adherence, or compliance, to the mindfulness intervention was the primary variable in this study. Therefore, the main independent variable was operationally different than a week-long mindfulness intervention, rather it was a dose-dependent variable. Importantly, there was no post-analysis to determine if mindfulness actually increased from this intervention, so no validity/manipulation check was performed to measure the outcome itself of the mindfulness intervention (increased mindfulness). Therefore, the main predictor of outcomes may not be specific to mindfulness, per se, but rather the individual basis of whether or not a participant is compliant. This issue is further nuanced by the differential finding in that those who adhered to the training on the first day performed better in the TMA task. Overall, is the cognitive performance enhancement more related to compliance, or mindfulness? Can this be parsed out? Is this a spurious relationship? This needs to be further considered and discussed. Other comments are below.

Intro

The authors do a nice job of pointing out that mindfulness has not consistently been found to increase cognitive performance, and suggest this is due to individual differences. I suggest the authors also include a discussion of how mindfulness itself as a variable may contribute to these differences, and discuss in which context mindfulness may be more likely to influence cognitive outcomes. For example, did these studies look at mindfulness as a trait, a brief mindfulness session, long term mindfulness training, etc? This is important as the current study implements a somewhat ‘in-the-middle’ approach between an acute session and long-term training.

The understanding of direction of alpha and theta activity are fundamentally different, so please explain in the broader sense what this statement implies/means: …”increases in alpha and theta oscillatory activity (see [50] for review).” – line 2, pg 6

This sentence is incomplete: “from mindfulness practice, as well as distinguishing “ – line 7 page 6

Page 6, please explain alpha in more detail (it’s stated generically as “alpha band” – explain what alpha band is and what this means). Similarly, alpha power and alpha frequency are not the same but reads as if they are in this paragraph (2 on page 6) of the paper. This suggestion follows true for other frequency bands mentioned in the intro (theta, gamma, and beta).

Line 28 on page 6 needs revision: “…various experimentation…”

Page 7: I’m not sure if this is accurately described as a theory? “..possible that steeper 1/f slopes may be observed long-term following brief/sustained mindfulness practice. Consistent with this theory, …”

Line 13 on page 7 says “practitioners”, but I think you don’t mean this word (a in a clinician?)?

Another possible incorrect word choice used for “equivocal” in line 23 of page 7.

Please explain what you mean here; “…in the effort to provide more nuanced findings.” Is there a benefit to providing more nuanced findings? This would suggest you are trying to find more variability, more complexities, as opposed to getting more directly at the relationship you are investigating.

You bring up low dimensional cognitive tasks, what does this mean exactly? The examples provided are mostly examples of executive functioning tasks, which would assume to translate to real-world applications. Please clarify.

What are ‘mutable’ tasks? Line 2 page 8.

In page 8, findings by Brown et al are the rational for an exploratory analysis, but this study is not mentioned prior to this statement. In general, the summary/concluding paragraph(s) of the introduction provides new content that was not previously supported or provided as rational for the purpose or hypotheses of the current study.

Please revise sentences throughout for word choice/meaning/clarity. For example, a hypothesis can’t be “determined” (a hypothesis is stated, not determined) that variables “may” (suggestive wording is not correct in a hypothesis) interact.

Methods

How many participants did you have to exclude if they reported history of neurological, cognitive, psychiatric or language disorders? Also, in the discussion or limitations section it would help to speak to the generalizability of this sample. It would possibly help also to be specific on what the inclusionary criteria were, and specifically the exclusionary criteria. I’m assuming some of this was exclusionary, as the recruitment aspect particularly did not bias participant recruitment by including reference to meditation or mindfulness.

For the sample size, please justify the power of this sample for statistical analyses. It appears to be underpowered.

What exactly was the personality questionnaire? Was this a validated measure of the personality traits you measured? If not the issue of construct validity needs to be addressed.

For measures, were there any missing data for any of the items in each of the surveys/questionnaires and if so how was this missing data addressed since the questionnaires were summed for a total score? Following this same notion, were there any participants that did not complete a specific portion of the study?

During the mindfulness intervention and protocol descriptions, please provide a little more detail. For example, could participants do all three 5-minute mindfulness sessions in a row, or did they have to be separated by a certain amount of time? Also, it is stated there were seven distinct recordings, what determined the recordings that were chosen? Did it matter if they listened to the same one each time? Additionally, was the same type of mindfulness implemented in each recording, or was this variable? How exactly were participants encouraged to complete the post-training questionnaire? Did this occur each day, only at the beginning, etc?

Since participants did not complete the same number of post-training surveys, and therefore do not have the same number of responses for the variable of enjoyment, what was the variability of responses? Likewise, please comment and provide inter-trial reliability of responses across sessions so that we can have a better understanding of how this statistical difference in averages regarding the denominator was calculated (e.g. some wouldn’t even have an average if they only responded once, while another person who responded all days would have a more accurate mean statistic).

Also, please clarify what you mean by the last sentence in paragraph 1 on page 11. It sounds like there would have been a max total of 7 responses per participant, but the range is 1 – 9. I am not clear on what this range therefore pertains to.

In EEG data analysis, please provide more details. For example, there is no mention of artifact rejection, impedance levels, how were eye blinks handled, how were right/left eye movements measured and handled, etc? Also, please provide rationale for high pass filter of 1 Hz. It was recently suggested this filter setting not go above .9 Hz for most studies, as the noise ratio greatly increase with higher filtering setting (below), so please address this.

(Zhang G, Garrett DR, Luck SJ. Optimal Filters for ERP Research II: Recommended Settings for Seven Common ERP Components. bioRxiv [Preprint]. 2023 Jun 14:2023.06.13.544794. doi: 10.1101/2023.06.13.544794. PMID: 37397984; PMCID: PMC10312706.)

How many people dropped out between time point one and two? It was stated 2 participants dropped out, so is this the attrition between time points?

Results

Figure 2 mentions data was log transformed, as typical in power analyses, but this is not mentioned in the methods.

Please expand on the analyses used from Corcoran et al (2018).

Why were occipital-posterior electrodes used for measure of interest? Typically frontal electrodes are used when measuring cognitive-related activity.

This model is underpowered with only 40 participants: = 0 + 1 * 2h * 3 + 4 + 5 + 0 + 

Some participants’ performance decreased in the TMA task at time point 2 – was this group analyzed separately and compared to those that increased their performance? Wondering if there is a fundamental difference between these participants. Also, did those that decrease their performance have a difference in their adherence?

Discussion

Overall this section is well done. Conclusions would be addressed pertaining to the initial paragraph of this review.

6. PLOS authors have the option to publish the peer review history of their article (what does this mean?). If published, this will include your full peer review and any attached files.

Reviewer #1: **Yes: **Eric Lopez-Maya

Reviewer #2: No

---

## [Author Response · Author response to Decision Letter 0]

5 Mar 2024

We thank the reviewers for the thorough evaluation of our manuscript and for the constructive and valuable feedback provided. Each comment has been carefully considered and significant changes have been made to the manuscript. To summarise critical changes, we have:

1) Added additional details of the mindfulness intervention and EEG pre-processing procedures 

2) Justified sample choice (pertaining to size, age range and exclusion criteria) 

3) Addressed copyright and file naming convention concerns 

We believe that the comments and associated changes have considerably improved the manuscript. Within our latest submission, we provide our response to the reviewers’ feedback and indicate where changes have been made to the manuscript.

---

## [Decision Letter · Decision Letter 1]

3 Apr 2024

PONE-D-23-29669R1Augmenting complex and dynamic performance through mindfulness-based cognitive training: an evaluation of training adherence, trait mindfulness, personality and resting-state EEG.PLOS ONE

Dear Dr. Dziego,

Thank you for submitting your manuscript to PLOS ONE. After careful consideration, we feel that it has merit but does not fully meet PLOS ONE’s publication criteria as it currently stands. Therefore, we invite you to submit a revised version of the manuscript that addresses the points raised during the review process. **It is recommended to complete the instructions indicated by Reviewer-2 so that the article is accepted.**

We look forward to receiving your revised manuscript.

Kind regards,

Thalia Fernandez, Ph.D.

Academic Editor

PLOS ONE

Journal Requirements:

Additional Editor Comments:

Please complete the Reviewer-2 issues so that the article can be accepted

Reviewers' comments:

Reviewer's Responses to Questions

**Comments to the Author**

1. If the authors have adequately addressed your comments raised in a previous round of review and you feel that this manuscript is now acceptable for publication, you may indicate that here to bypass the “Comments to the Author” section, enter your conflict of interest statement in the “Confidential to Editor” section, and submit your "Accept" recommendation.

Reviewer #1: (No Response)

Reviewer #2: (No Response)

2. Is the manuscript technically sound, and do the data support the conclusions?

Reviewer #1: (No Response)

Reviewer #2: Yes

3. Has the statistical analysis been performed appropriately and rigorously? 

Reviewer #1: (No Response)

Reviewer #2: Yes

4. Have the authors made all data underlying the findings in their manuscript fully available?

Reviewer #1: (No Response)

Reviewer #2: Yes

5. Is the manuscript presented in an intelligible fashion and written in standard English?

Reviewer #1: (No Response)

Reviewer #2: Yes

6. Review Comments to the Author

**Reviewer #1:** (No Response)

**Reviewer #2: **Thank you for thoroughly addressing my previous comments. The manuscript overall is much improved and considered acceptable for distribution. Only minor comments remain:

1.Wording in abstract sounds like enhancements were seen in resting state EEG when no changes were observed pre and post intervention. Suggest revision to make more clear.

2. Please remove language from correlational analyses that imply causation (e.g. sub header “effect of individual personality traits” – personality traits were not manipulated. Causational language is seen in other examples throughout text). I understand that the authors are referring to the main effects from each analysis, but within the discussion this language suggests the interpretation of the relationship as causal.

7. PLOS authors have the option to publish the peer review history of their article (what does this mean?). If published, this will include your full peer review and any attached files.

Reviewer #1: No

Reviewer #2: No

---

## [Author Response · Author response to Decision Letter 1]

11 Apr 2024

We thank the Editor and Reviewers for their final minor comments on our manuscript. Each comment has been carefully considered and amended in the manuscript to suit. We look forward to receiving any further feedback on our responses and revisions.

---

## [Editor Report · Decision Letter 2]

6 May 2024

Augmenting complex and dynamic performance through mindfulness-based cognitive training: an evaluation of training adherence, trait mindfulness, personality and resting-state EEG.

PONE-D-23-29669R2

Dear Dr. Dziego,

We’re pleased to inform you that your manuscript has been judged scientifically suitable for publication and will be formally accepted for publication once it meets all outstanding technical requirements.

Kind regards,

Thalia Fernandez, Ph.D.

Academic Editor

PLOS ONE
---

## [Editor Report · Acceptance letter]

9 May 2024

PONE-D-23-29669R2 

PLOS ONE

Dear Dr. Dziego, 

I'm pleased to inform you that your manuscript has been deemed suitable for publication in PLOS ONE. Congratulations! Your manuscript is now being handed over to our production team.

Kind regards, 

on behalf of

Dr. Thalia Fernandez 

Academic Editor

PLOS ONE